# Correction of Substrate Spectral Distortion in Hyper-Spectral Imaging by Neural Network for Blood Stain Characterization

**DOI:** 10.3390/s22197311

**Published:** 2022-09-27

**Authors:** Nicola Giulietti, Silvia Discepolo, Paolo Castellini, Milena Martarelli

**Affiliations:** 1Department of Mechanical Engineering, Politecnico di Milano, 20156 Milano, Italy; 2Department of Industrial Engineering and Mathematical Science, Università Politecnica delle Marche, 60131 Ancona, Italy

**Keywords:** blood stains, hyper-spectral imaging, neural network, forensics

## Abstract

In the recent past, hyper-spectral imaging has found widespread application in forensic science, performing both geometric characterization of biological traces and trace classification by exploiting their spectral emission. Methods proposed in the literature for blood stain analysis have been shown to be effectively limited to collaborative surfaces. This proves to be restrictive in real-case scenarios. The problem of the substrate material and color is then still an open issue for blood stain analysis. This paper presents a novel method for blood spectra correction when contaminated by the influence of the substrate, exploiting a neural network-based approach. Blood stains hyper-spectral images deposited on 12 different substrates for 12 days at regular intervals were acquired via a hyper-spectral camera. The data collected were used to train and test the developed neural network model. Starting from the spectra of a blood stain deposited in a generic substrate, the algorithm at first recognizes whether it is blood or not, then allows to obtain the spectra that the same blood stain, at the same time, would have on a reference white substrate with a mean absolute percentage error of 1.11%. Uncertainty analysis has also been performed by comparing the ground truth reflectance spectra with the predicted ones by the neural model.

## 1. Introduction

Hyper-spectral imaging (HSI) is an emerging technique with a quite wide range of applications, from food to medicine, from quality control to forensics. This technique provides spatially resolved spectra, also called hyper-cube, with two spatial dimensions and one spectral dimension. In [1], Lu et al. presented a review of hyper-spectral imaging technologies and their applications in the medical field for tissue physiology and morphology characterization and composition assessment. Hyper-spectral cameras have also been exploited in the forensic sector for enabling the forensic analysts to perform both the geometric characterization of biological traces (fingerprints minutiae extrapolation, blood trace shape, etc.) and the trace classification based on the chemical composition a spectral emission is related to. Indalecio-Cespedes et al. [2] stressed the importance of the substrate on blood stain visualization. Matthew E. James [3] explored the use of an alternate light source to acquire blood stain hyper-spectral images on dark and/or patterned fabrics. In Majda et al. [4], hyper-spectral cubes were treated with chemometric methods (minimum noise fraction and principal component analysis), leading to a rapid analysis of the homogeneity of samples, selection of areas, and following the changes occurring in biological traces. The main limit of the analysis is related to the use of the Dry Blood Spot (DBS) samples, which is not a representative case in real applications. In Edelman et al. [5], a comprehensive review on hyper-spectral imaging for non-contact analysis of forensic traces was presented. This paper provides an overview of the theory, instruments, and processing techniques involved in hyper-spectral imaging, explaining recent advances in forensic science applications. An important factor that is considered is the interaction between light and the sample, which is related to the optical properties of the latter. However, only the sample layer has been considered, while in the case of a blood stain deposited on a substrate, the conceptual model of the sample has to be extended taking into account the effect of all different layers (Figure 1), including the substrate. At first, the light interacts with the outer surface of the blood sample. This is related to reflection and diffusion, and gives usually a limited amount of information about the sample. Then, the light interacts with the blood itself. In this case, both diffusion and absorption phenomena occur, with light scattered by blood particles. Consequently, the light interacts with the substrate, which has not been considered in Edelman et al. [5]. The portion of light transmitted in the blood interacts with the substrate surface via reflection, scattering, and transmission. Finally, the remaining portion of the transmitted light interacts with the sample material and is scattered back through the substrate and blood stain towards the camera.

All those effects are wavelength-dependent. Absorption in the visible wavelength, Near Infra Red (NIR) and Infra Red (IR) range correspond to different behavior of the molecules of the material, and all can be exploited to identify the chemical contents of a specimen through the hyper-spectral camera. In Edelman et al., it is noted that the analysis of traces on substrate encountered in casework is one of the most important challenges for future works. This issue has been studied in Edelman et al. [6], where the authors demonstrated that blood stain analysis is feasible on white substrate but still presents some problems with colored substrate, mainly if they are dark.

The problem of substrate material and color becomes a serious issue in a real crime scene, which totally differs from an ideal laboratory condition. In this case, it becomes tricky to distinguish blood from other stains and then analyze it. Artificial Intelligence-based (AI) approaches have been extensively applied in the forensic field to examine blood samples. Many authors, given the relevance of discriminating blood stains from other substances, in any material, in a non-invasive way, exploited AI to classify one blood stain against drops of other substances. In Zulfiqar et al. [7], hyper-spectral images acquired on a total of 225 blood (different donors) and non-blood samples have been analyzed using five different classifiers: Support Vector Machine (SVM), Artificial Neural Network (ANN), k-nearest neighbor Neural Network (KNN), Random Forest, and Decision Tree. Ksizaek et al. [8] classified blood stains from other substances such as tomato sauce, ketchup, and artificial blood employing different neural network models such as 1D, 2D, and 3D Convolutional Neural Networks (CNNs), Recurrent Neural Networks (RNNs), and a Multi-Layer Perceptron (MLP). Then, the high performance of those methods are compared with the baseline results of SVM. In Palka et al. [9], Genetic Algorithms (GA) are used for pattern and band selection in the classification of hyper-spectral images characterized by both blood stain and non-blood stains. In this trial, they demonstrated that the GA-based model succeeds in decreasing the number of bands and creates an accurate classifier. Yang et al. [10] showed that spectral features can be used for blood stain detection in crime scene backgrounds. They give also evidence that blood spectral signatures contain unique features related to persons involved or to the nature of the violence, but they are heavily influenced by the substrate surfaces where the stain has been dropped off. Non-blood stains often mislead the detection and can generate false positives in a real crime scene, especially for dark and red substrates. The main features used to differentiate between blood and non-blood samples are the peaks and depths of the reflectance spectra. In order to support system design and parameter selection, in Kerekes et al. [11], an analytical end-to-end model that forecasts remote sensing system performance is developed. The model considers three main components (the scene, the sensor, and the processing algorithms) and applies statistical descriptions of the scene class reflectances. A similar approach, even if it is very useful for the system design, parameter trade-off, and the understanding of phenomena, is in practice not suitable to compensate for errors linked to substrate influence, for example, to cope with the uncertainty typical of real measurements.

The present work presents a method to identify blood stains and correct their reflectance spectra from the influence of substrate. The procedure is then based in two phases. The first one consists in the recognition of blood traces from other substances. This is done by means of an MLP network, a fully connected class of ANN that performs a binary classification of the collected spectra from the different substances. As mentioned in Ksizaek et al. [8], an MLP network achieves competitive results in blood stain classification despite its simple architecture also in comparison with more sophisticated Deep Neural Network (DNN) models (e.g., CNN). The use of MLP exploits very lightweight and fast models (on the order of 0.1 Mb) as opposed to advanced DNN models that are much heavier.

Then, upon the blood stain is detected, the second phase consists in the correction of the blood reflectance spectrum in order to get rid of the substrate influence. The correction is performed by using an MLP able to perform a blind blood-stain substrate correction. Starting from the reflectance spectra of a blood stain dropped off on any substrate, the algorithm allows to obtain the reflectance spectra that the same blood stain, at the same time, would have on a white reference material. Neglecting the substrate contribution, it is possible to analyze the absorption peaks of hemoglobin, such as the Soret peak (400 nm), the β and α absorption spectral bands (540 nm and 576 nm), and the change in their pattern considering the time course, once transformed into met−Hb and HC [12,13]. These features, which would otherwise be lost, are proven to be critical in forensic investigations and analysis for blood traces recognition or blood stain age estimation [4,14].

The structure of the paper is as follows: Section 2 describes the hyper-spectral images acquisition procedure and Section 3 the image processing methods based on MLP for the blood stain binary classification and based on MLP for substrate distortion correction. Section 4 presents the results of the blood stains identification and substrate correction and the analysis of the uncertainty inherent to the correction method itself. The models are finally tested on blood stains deposited on materials different from the ones used for the network training.

## 2. Material and Methods

Establishing the substrate influence on the blood stain in hyper-spectral imaging may prove difficult. In fact, the substrate contribution besides depending on type of material and texture, strongly relates on the light illumination source and on the blood composition, which varies over time. Fresh blood deposited as a stain on a surface is a complex suspension of cellular components, consisting of red blood cells, white blood cells, platelets, and plasma. Plasma is composed of water, proteins, inorganic electrolytes, and clotting factors. Over time, the cellular components and plasma play different roles in the formation of the blood model [15]. At the time of non-uniform evaporation and subsequent drying of the blood stain, three regions are generated: a dried region (solid state), a saturated region (gelled state), and a supersaturated region (liquid state). As the liquid evaporates, plasma gelation occurs at the edge of the drop due to the increased concentration of protein macro-molecules, while inorganic electrolytes are pushed into the central part. Subsequent evaporation leads to the generation of the cracking pattern [15]. The system described in this paper has been optimized in terms of light efficiency, from the hardware side, and has been tested for different substrates and different deposition times of the stain. In this section, the hyper-spectral imaging setup and acquisition procedure will be described first, and subsequently, the image processing strategy will be detailed.

### 2.1. Test Bench

The hyper-spectral system used for the collection of blood stain samples deposited on different substrates is the Hinalea 4250 featuring a tunable filter that, placed in front of the sensor, sequentially selects the visible and NIR spectral bands in an extended range between 400 nm and 1000 nm, with a step size of 2 nm (300 nominal spectral bands), generating a hyper-spectral cube. The hyper-spectral cube is characterized by a spatial dimension of 2.3 MP (RGB sensor effective monochromatic equivalent 588,544 pixel without demosaicing). The images are acquired at a fixed distance of 30 cm from the sensor, obtaining a spatial resolution of 0.06 mm/pixel.

The camera is set on top of the blood stain sample in between of two halogen lamps, as shown in Figure 2. The use of halogen lamps provides uniform, broadband illumination, which is essential to ensure the right light intensity over a wavelength range as wide as possible. The choice of the light properties in terms of spectral content (broadband vs. narrow-band), directional distribution (directional vs. diffuse), and uniformity over the sample surface, is crucial for the accuracy of the image registration. The spectral analysis is limited to the range between 412 and 774 nm, with a step size of 2 nm (185 nominal spectral bands), since the maximum spectral intensity of halogen lamps falls within this range. As explained in Section 1, this interval includes characteristic features of blood.

### 2.2. Image Acquisition

Blood given by a healthy female volunteer is collected in acid vacutainer containers and it is deposited with her consent, through a pipette, directly on diverse substrates. The size of the blood stain deposited in the respective substrate is kept to approximately one centimeter in diameter. Obviously, this condition, although performed with the upmost precision, varied according to the blood absorption in the target substrate. Twelve substrates of common use with different colors and textures are chosen. In particular, from completely absorbent materials, e.g., light jeans, dark jeans, red fabric, green sponge, wood, and napkin, to materials such as white paper, red paper, black paper, yellow paper, and finally, a white ceramic tile. A wide range of material is chosen in order to analyze blood stains not only in light-colored materials, where the latter one is very visible even to the naked eye, but also in other substrates where its influence can degrade and blend in with the blood spectra. For instance, the red substrates can modify the reflectance blood spectra due to its color that alters the characteristic peaks of hemoglobin, or the dark substrate where the stain is even not noticeable. Blood stains are deposited on rectangular segments of substrate measuring 5.25 cm × 3.25 cm. A total of 12 samples of blood stains on the different substrates will be analyzed. Each sample is identified by an ID as reported in Table 1. As shown in Figure 3, the samples are grouped into 2 × 2 arrays such that they occupy the entire field of view of the camera at the minimum focus distance of the optical system.

Prior to each hyper-spectral cube acquisition, a dark and flat field-calibration procedure is performed for reflectance measurements. The flat field calibration is exploiting framing images of a reference white target exposed to the test bench. The blood stain deposited on the white ceramic tile, identified by $C_{5}$, is assumed as a reference sample for the analysis of the hyper-spectral cubes. This material is selected for two main reasons:The white light color provides proper visualization of the blood stain without interference from the substrate color.Its non-absorbent behavior makes it possible to have no distinction between the reaction that the blood may have on the substrate surface and on the substrate portion where it is absorbed [16].

In order to collect data for the training of an ANN capable of recognizing pixel belonging to blood and non-blood stains, four separate substrates of pink cotton, blue cotton, white cotton, and light wood are prepared on which soy and ketchup stains are deposited. Figure 4 shows how visually the stains of different materials may look very similar to each other (a), but by analyzing the average reflectance spectra, the blood retains its typical, clearly recognizable characteristics (b).

Hyper-spectral cubes are framed once a day for 12 days for each sample. At day one, the acquisition is performed every hour for eight hours. The continuous acquisition of the first day is conducted with the purpose of estimating dynamic spectral changes due to hemoglobin oxidation, while the prolonged acquisition with the intention of seeing spectral alterations, although minor, due to blood-aging [12].

## 3. Image Processing and Data Collection

For each acquired hyper-spectral image, the area belonging to the stains (blood and non-blood stains), and the area belonging to the substrate, are manually selected with a circular ROI (radius = 50 pixel). Every single reflectance spectrum is treated as a single observation. Each observation is labeled according to the substrate identification name (Table 1), acquisition time, and marked as blood pixel or not-blood pixel (e.g., soy/ketchup pixel or substrate pixel). These spectra are then normalized with respect to the maximum reflectance value recorded in the reference white tile substrate $C_{5}$ at time t=0 h.

### 3.1. Blood Stain Spectra Binary Classification via Neural Networks

Before creating a model capable of correcting the effect of the substrate on blood deposited on a surface, it is necessary to be able to distinguish spectra belonging to a blood stain from spectra belonging to other substances, even those that are visually very similar. An MLP network is exploited to perform a binary classification of the collected spectra. MLP has been used for a long time in numerous scientific and technological sectors and has proven to be able to solve very complex problems in several fields [17]. This model architecture is able to predict new continuous outputs from new statistically independent input data starting from a set of continuous input–output variables. The artificial neuron is the basic element of an MLP. This is a module that sums weighted inputs and passes them through a cost function that determines its output. One or more neurons are arranged into layers of neurons and several layers are arranged into a network of layers, also called neural network. The selection of the activation functions, the number of layers, and the number of neurons for each layer, depend on the complexity and the nature of the problem under examination. The weights of the model are trained by a back-propagation algorithm, typically exploited in a supervised learning technique [18,19,20]. The model proposed in this paper is shown in Figure 5. The ANN takes reflectance spectra as input, and as output, returns a single value *b*: 1 if the spectrum belongs to a blood stain, 0 in all other cases.

MLP trains using stochastic gradient descent optimization algorithms that change the parameters of the model, such as weights and learning rate, in order to increase model performance [21]. This involves having to choose an error function, also called loss function, which estimates the error of the model in order to update weights to reduce the error iteratively. As this is a regression problem, the binary cross-entropy is used as loss function [22]. From the spectra acquired in Section 3, 200,000 spectra are randomly selected from blood stains deposited on different substrates and with different ageing time, and 200,000 spectra not belonging to blood stains (e.g., substrates or threshold/ketchup stains). The acquired dataset was split into three parts: train, validation, and test set with a split ratio of 8:1:1. The first two datasets were used in the training and optimization phase of the model, while the last one was used to test the performance of the output model. The number of layers, neurons, and hyper-parameters were tuned according to a Bayesian optimization approach. This approach, unlike the more classical methods of grid search and randomized search, has proved to be advanced and faster in hyper-parameters optimization tasks for neural networks [23,24,25,26]. A Bayesian optimization technique performs the training many times with different sets of number of layers, neurons, and hyper-parameters. The method initially tests the function for *n* times by assigning causal values to its variables in order to diversify the exploration space. Subsequently, based on the values obtained in the previous step, the actual Bayesian optimization begins. At each step, the algorithm evaluates the past model information to select new optimal parameter values that increase newer model performances according to the method described in [27,28]. The first step is the definition of the function to be optimized. Given the nature of the problem, the objective function is constituted by the neural network model, which takes as input several parameters, and has the binary cross-entropy as its output. The precise purpose of the optimizer is to maximize the value of this coefficient. Once the function to be optimized has been chosen, it is necessary to determine which input parameters have to be optimized and their corresponding bounds, since this is a constrained optimization method. The input parameters of the function that are made to vary in such a way as to minimize the binary cross-entropy are:Batch size bs, a hyper-parameter that defines the number of training data sub-samples that will be propagated through the network. The batch size was varied between 256 and 2048.Learning rate lr, a hyper-parameter that defines how much the model will change in response to the estimated loss each time the model weights are updated by the optimizer. The learning rate was varied between 0.0001 and 0.1.Number of epochs *e*, a hyper-parameter that defines the number of times a whole dataset is passed through the neural network model. The number of epochs was varied between 20 and 200.Activation function af, a hyper-parameter that defines the relation between the weighted sum of the input and the output from an artificial neuron or from the set of artificial neurons included in a layer of the network. The activation function can be selected among linear, Sigmoid, ReLU, and Tanh [29].Optimizer *o* defines the algorithm exploited to reduce the loss function-modifying attributes of the neural network, such as weights and learning rate. The optimizer can be selected among SGD, Adam, RMSprop, Adadelta, and Adagrad [21].Number of layers nl ranging between 1 and 12.Number of neurons per layers nn ranging between 10 and 400.

The objective function was thus defined by the following equation:(1)BinaryCrossEntropy=f(x,y,bs,lr,e,af,o,nl,nn)
(2)BinaryCrossEntropy=−(p(x)log(y)+(1−p(x))log(1−y))
where *x* is the input spectrum, *y* is the output prediction (from 0 to 1), and *p*(*x*) the ground truth (1 or 0). The model, once trained, is able to predict whether or not the spectra under examination belong to a blood stain.

### 3.2. Substrate Distortion Correction via Neural Networks

In the previous section, an ANN model capable of classifying a spectrum as blood or non-blood was realized. Following the same concepts, a model capable of correcting the substrate spectral distortion in hyper-spectral blood stain images was trained. In this case, the MLP network involved a regression to predict a real-value quantity, specifically the reflectance spectra corrected by the substrate influence. The model proposed in this task is shown in Figure 6; the model takes the blood stain reflectance spectra on a substrate $C_{n}$ at time $t_{i}$ as input, and outputs the spectra the blood stain would have on a white tile reference substrate $C_{5}$ at the same time $t_{i}$.

As this is a regression problem, the coefficient of determination $R^{2}$ was used as loss function because it is proven to be more informative than other indicators used in regression analysis evaluation [30]. The coefficient $R^{2}$ was calculated according to Equation (Equation 3). $\lambda_{i}$ represents the observed dataset (ground truth), *n* the size of the dataset, and $\lambda_{i}^{*}$ the predicted values. $\bar{\lambda }$ is the average of the observed data and it is used for the calculation of the total sum of squares. The value of $R^{2}$ varies between −∞ and 1. When it is equal to 1, the model represents the data perfectly. When $R^{2}$ assumes negative values, it means that the model does not follow the trend of the data [31].
(3)$$R^{2}=1-\frac{\sum_{i=1}^{n}{\left(\lambda_{i}-\lambda_{i}^{*}\right)}^{2}}{\sum_{i=1}^{n}{\left(\lambda_{i}-\bar{\lambda }\right)}^{2}}$$

From the spectra acquired in Section 3, 380,000 blood stain spectra were selected with known deposition substrate and ageing time. The acquired dataset was split into three parts: train, validation and test set with a split ratio of 8:1:1. The model was trained exploiting the Bayesian optimization technique, already discussed in Section 3.1, varying only the target objective function.

### 3.3. Procedure Workflow

The procedure workflow for the blood stain identification and for the correction of substrate spectral distortion in the blood reflectance spectra is shown in Figure 7 and can be described as follows.

aThe job starts by acquiring a hyper-spectral image containing the blood stain to be analyzed according to the procedure described in Section 2.1 and Section 2.2.bThe Region Of Interest (ROI) containing the blood stain is selected, resulting in *N* reflectance spectra, where *N* is the total number of pixel in the ROI.cThe first reflectance spectrum is used as input by the binary classification model of which the optimal weights were obtained during the training phase of the model, as described in Section 3.1.dThe model verifies that the spectrum belongs to a blood stain. If the model considers the current spectrum as not belonging to a blood stain, it is discarded and the procedure is repeated from step “a” with the next spectrum. If, on the other hand, the spectrum is classified as blood spectrum, then this is used as input to the inferential model that deals with the removal of the background spectral distortion.eThe optimal weights and model architecture are obtained according the procedure described in Section 3.2. The output spectrum is stored in the memory and the procedure is iterated from step 3 for each available reflectance spectrum.fFinally, the average corrected spectrum is returned as output and can be used for extraction of specific parameters, such as the blood age, for example.

## 4. Analysis of Results

This section is organized as follows. First, the performance obtained from training the two proposed AI models is described. Next, the spectra acquired and the effect of the removal of the substrate contribution has on them will be analyzed. Subsequently, an analysis of the model accuracy on the reconstruction of the blood spectra corrected from the substrate influence is presented. Finally, the proposed method is tested on blood stains deposited on two materials different from the ones used for the training. The chosen materials are brown and white cotton fabric.

### 4.1. Blood Stain Spectra Binary Classification Model

Following the Bayesian optimization of the model’s hyper-parameters, described in Section 3.1, the model is trained with the training and validation dataset. A stratified k-fold cross-validation [32] with 10 fold is performed, obtaining a mean classification accuracy of 98.9% with a standard deviation of ±0.5% on the test dataset. This seems consistent with results in [8,9]. The parameters selected for the optimization are:Batch size bs=1353.Learning rate lr=0.0436.Number of epochs e=76.Activation function af=linear.Optimizer o=Adam.Number of layers nl=4.Number of neurons per layers nn=[111,105,54,17].

Due to the very high accuracy achieved, it is possible to use this model upstream of the substrate influence correction model. In this way, only those pixel believed to belong to a blood stain are processed. From the analysis of the trend of loss function as epochs increase, the model does not appear to be affected by over-fitting showing the same accuracy on training, validation, and test dataset. In fact, comparing the results in the training, validation, and test datasets, yields an accuracy of 98.9, 98.6 and 98.5, respectively.

### 4.2. Substrate Influence Correction Model

The neural network model for substrate influence correction is trained through the Bayesian optimization process described in Section 3.2. The resulting model that maximizes $R^{2}$ is composed by 5 inner layers (nl) with, respectively, 407, 201, 120, 78, 11 neurons (nn). The optimal activation functions (af) selected are ReLU for the inner layers, and linear for the output layer. The other parameters, which guarantee the highest $R^{2}$ value, are the following:Batch size bs=517.Learning rate lr=0.0001.Number of epochs e=91.Optimizer o=Adam.

The model is then trained with a train and a validation dataset collected as described in Section 3.2. Finally, the model is cross-validated with the k-fold method [32] (10 folds) on the test dataset, giving an $R^{2}$ value of 0.87 ± 0.02 A.U. and a mean absolute error of 0.041 ± 0.05 A.U. From the analysis of the trend of loss function as epochs increase, the model does not appear to be affected by over-fitting, showing the similar $R^{2}$ and mean absolute error on training, validation and test datasets. In fact, comparing the results in the training, validation and test datasets yields an $R^{2}$ mean value of 0.87, 0.86, and 0.86, respectively.

### 4.3. Analysis of Acquired Spectra

Figure 8 shows the reflectance spectra of blood stains on the reference white tile substrate $C_{5}$ in the visible and NIR range. The blue curve represents the reflectance spectrum acquired at time t=0.5 h where the typical blood features are visible, i.e., the absorption peaks of $HbO_{2}$, at 416 nm (the Soret peak), 542 nm (the β band) and 576 nm (the α band), and the characteristic narrow slope of fresh blood in the spectral band extended between 600 nm and 650 nm. The orange curve, instead, is the reflectance spectrum acquired at time t=288 h. In this case, it is possible to observe significant differences in blood spectra, not only related to the decrease of α and β peak intensities, but also a softer transition from $HbO_{2}$ to met−Hb. As clearly visible at 620 nm, the curve changes convexity and its slope decreases.

Figure 9 shows the blood spectra for different substrates; samples have been acquired at time t=0.5 h. Each reflectance spectrum is obtained as the average spectrum of all pixels belonging to the blood stain on same substrate $C_{n}$ at the same time $t_{i}$. From here on, we will refer to average spectra. Observing the curves in Figure 9, it becomes evident why substrate removal is a major challenge; in fact, the detection of the blood stain characteristic features that allow blood age determination is complicated due to the substrate light absorption. For example, dark materials (such as the dark jeans, i.e., line green in Figure 9) show flat trends throughout the entire spectral range, making it difficult to discriminate blood features. In red materials, the α and β peaks disappear. Only the reflectance spectrum of blood deposited on the reference substrate, the white ceramic tile $C_{5}$, allows immediate identification of the blood characteristic features.

### 4.4. Analysis of Blood Spectra with Substrate Distortion Correction

The method proposed in Section 3.3 is applied to hyper-spectral images of blood stain deposited on wood and black paper at time t=50 h of ageing. Results are reported in Figure 10 and Figure 11, respectively. In the left plots of both figures, the blood spectrum in one pixel of the blood stain image is presented, while in the right plots, the average spectra obtained by averaging the spectra over all the pixels of the stain are presented. The blue line represents the input spectrum, the orange line is the reference white-tile spectrum, and the green line is the predicted corrected spectrum. Even starting from a reflectance spectrum that is greatly altered by non-collaborative substrates (e.g., wood or black paper), it is still possible to reconstruct the spectrum unaffected by the substrate (Figure 11). In the case of black paper, the blood stain, as shown in Figure 3, is not visible to the naked eye because it is completely covered by the color of the surface on which it is deposited. In both cases, it is evident how crucial it is to consider the average of all acquired spectra. In fact, the spatial averaging allows to get rid of the non-homogeneous sample texture which is a characteristic of typical materials considered in forensics. In the case of black paper, it can be noticed that the main deviation from the ground truth is in the wavelength range between 600 nm and 700 nm, as it will be confirmed in Section 4.5.

### 4.5. Neural Model Sensitivity Analysis

In this section, an analysis of the model accuracy on the reconstruction of the blood spectra corrected from the substrate influence is presented. The error in corrected spectrum prediction has been defined as the Mean Absolute Percentage Error (MAPE), obtained as the percentage difference between the predicted spectra achieved by the ANN model and the reference spectra of the blood stain on the reference substrate $C_{5}$, averaging all pixel spectra belonging to the blood region. It should be noted that the analysis was performed on samples of different materials and acquired at different times. Therefore, MAPE varies depending on the substrate, acquisition time, and spectral wavelength. Figure 12 illustrates the statistical distribution of the Mean Absolute Percentage Error, considering all the samples examined, i.e., all the substrates, all the acquisition times, and the entire spectral range collected. The distribution of the MAE is a zero-centered, almost Gaussian, with a standard deviation of σ = 1.13%. Figure 13 shows the MAPE in function of the spectral wavelength for the complete set of substrates and acquisition times. The MAPE is the central curve represented by the circles. The figure presents also the standard deviation, e.g., the vertical bars, which reveal that at low wavelengths, up to 450 nm, a larger standard deviation occurs. In this spectral region, as shown in Figure 14, the halogen-lamp used for illumination does not guarantee a sufficient spectral intensity, thus returning results affected by high level of noise. The uncertainty becomes more stable above 450 nm, with a second peak of deviation in the extended range between 600 nm and 650 nm. This is due to the dark-colored substrates, such as blue jeans and black cardboard, which absorb at these wavelengths, causing a greater variance. The deduction is evidenced if the MAPE and the related standard deviation σ are observed for the different samples, as given in Figure 15. It is evident that the greatest standard deviation is concentrated at the shortest wavelengths for all the substrates and that there is a high standard deviation in the wavelength range between 600 and 650 nm, mostly for the black paper, as clearly shown in Figure 11. The latter can be the outlier that increases the standard deviation in this range of wavelengths of all the samples, as evidenced in Figure 13.

The MAPE and standard deviation of each substrate are reported also in Figure 16, but averaged within all the spectral wavelengths. It is confirmed that the standard deviation is large in the case of dark colored materials, e.g., black paper and dark jeans. However, the black jeans substrate presents a MAPE lower than other substrates (Figure 15). This is because, due to the texture of the material, it suffers from greater variability, and when averaged, the effect is compensated for.

Since one of the other parameters considered in the study presented in this paper was the age of the blood stain, the MAPE has been analyzed as a function of the acquisition times of all the samples. Its trend is plotted in Figure 17. The average MAPE is represented by the circles, while the vertical bars represent the standard deviation. It can be observed that the MAPE and its variance are stable near to zero, and therefore, it is possible to conclude that the MAPE is independent from the blood stain age.

### 4.6. Test on New Materials

The method proposed is finally applied on blood stains deposited on two new materials different from the ones used for the network training: brown and white cotton. The acquisition is carried out at different times of blood ageing: 1, 24, 47, and 96 h. Again, blood deposited on a white tile is used as a reference. The results in terms of MAPE are shown in the Table 2; it is evident that the error is always below 5%, as it was for the materials used for the network training (Figure 17). As shown in Figure 18 and Figure 19, the method succeeds in reducing the substrate effect on reflectance spectra of blood stains deposited for 4 days also in complex substrates such as a brown cotton fabric (Figure 19).

## 5. Discussion and Conclusions

This work presents an AI-based method for reflectance-spectra substrate distortion correction on blood stains deposited on several kind of substrates. The application of this correction enables forensic analysts to evidence and analyze traces set down in a real crime scene with a high level of accuracy.

In particular, it has been demonstrated that, starting from a blood stain deposited on diverse substrates, it is possible to obtain the reflectance spectra that the same stain at the same time would have on a standard reference white substrate with a MAPE of 1.11% and a standard deviation of 1.13%. For this purpose, two AI-based models have been developed. The first neural network model estimates, with 98.9±0.5% of accuracy, whether the spectra under investigation belongs to a blood stain or not. The second one performs the background distortion correction with a MAE of 0.041±0.05 A.U. and a $R^{2}=0.87\pm 0.02$ A.U. Both networks were trained exploiting a hyper-parameter Bayesian optimization technique and evaluated with a k-fold cross-validation method with 10 folds. It has been shown how, through the use of simple, lightweight MLP models, the performance of more complex DNN architectures can be achieved, consistent with results in [8,9]. The results of this study showed that this new approach can achieve high performance for all substrates under investigation, being able to obtain a reflectance spectra traceable to the reference one regardless of the surfaces involved in a hypothetical crime scene under investigation. The proposed method was finally applied to blood stains deposited in two materials that were not used in the training phase of the model, resulting in a MAPE of less than 5% for all ageing times considered. It has been proved that the inaccuracy of the model is strictly related to the illumination source, since the highest level of uncertainty is concentrated on the wavelength where the illumination intensity is low. It can be deduced that the use of light sources capable of covering a wider spectral range towards the ultraviolet band could contribute strongly to the decrease of the standard deviation in that spectral band. This could be achieved through mixing multiple light sources. The results obtained in this work may allow the use of techniques known in the literature for estimating the age of blood deposited on any surface.

## Figures and Tables

**Figure 1 sensors-22-07311-f001:**
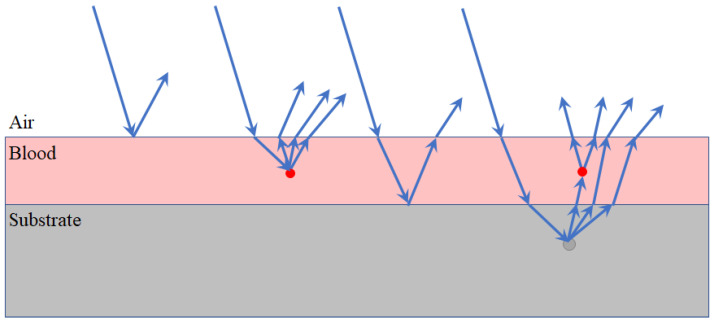
Scheme of the light scattering paths on blood stain over a substrate.

**Figure 2 sensors-22-07311-f002:**
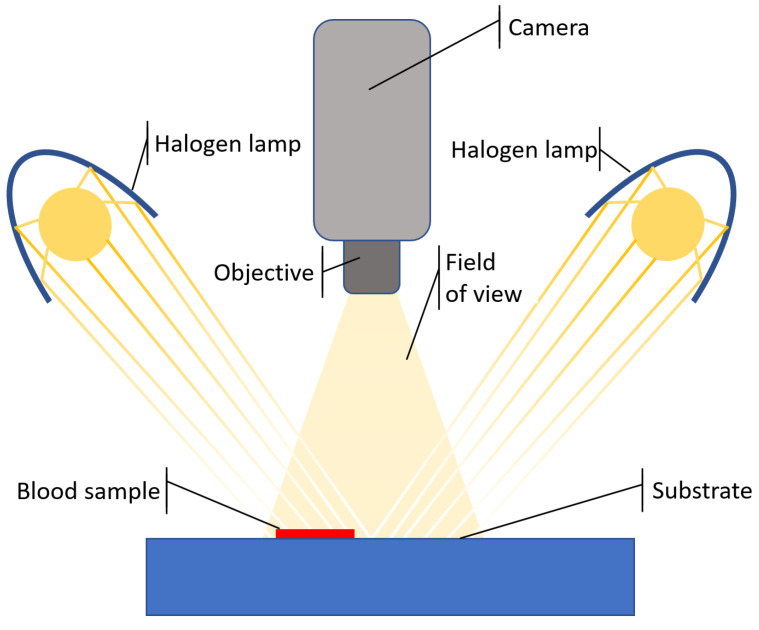
Hyper-spectral image acquisition setup scheme.

**Figure 3 sensors-22-07311-f003:**
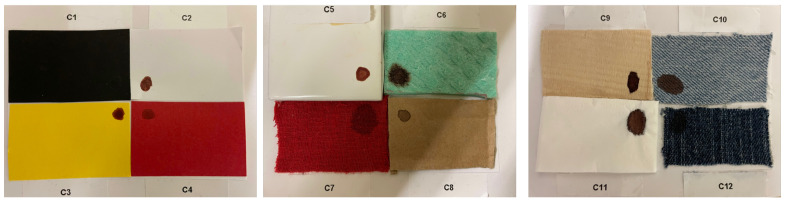
Blood stains samples on the 12 different substrates.

**Figure 4 sensors-22-07311-f004:**
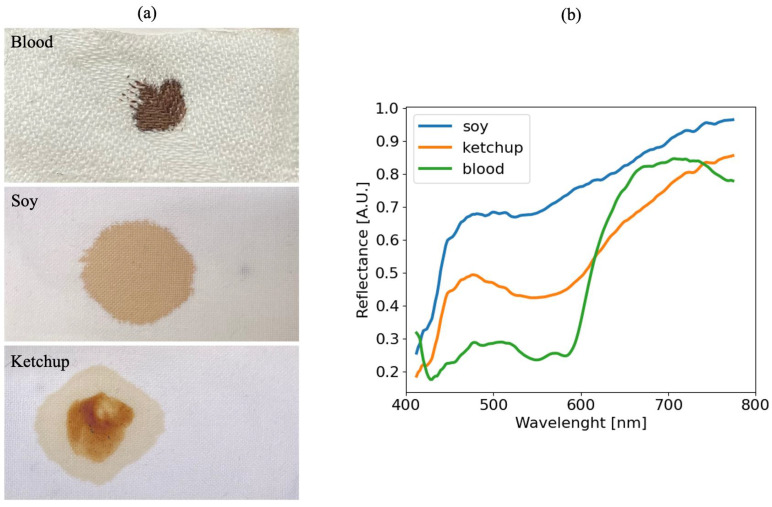
Blood, soy, and ketchup stain images (**a**), reflectance spectra comparison (**b**), [A.U.] means non-dimensional unit.

**Figure 5 sensors-22-07311-f005:**
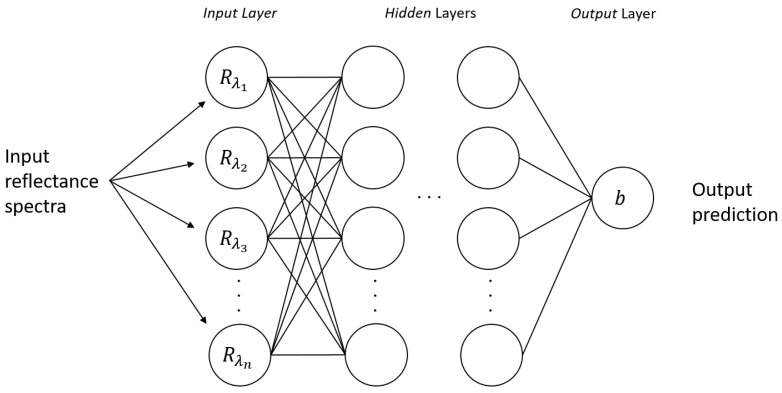
Multi-Layer Perceptron model for blood stain reflectance spectra binary classification.

**Figure 6 sensors-22-07311-f006:**
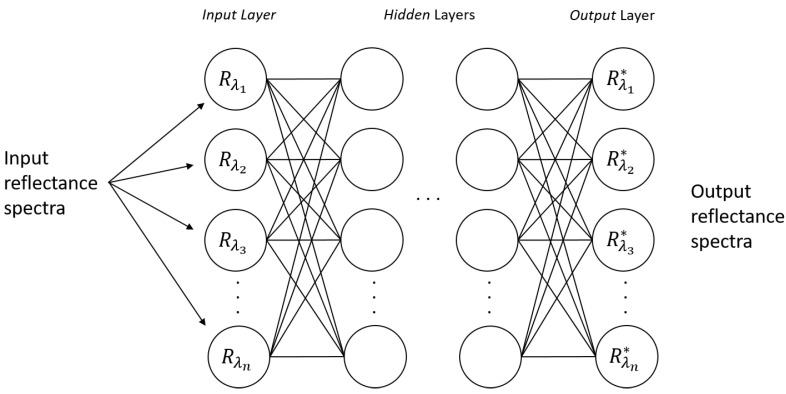
Multi-Layer Perceptron model for the correction of substrate spectral distortion.

**Figure 7 sensors-22-07311-f007:**
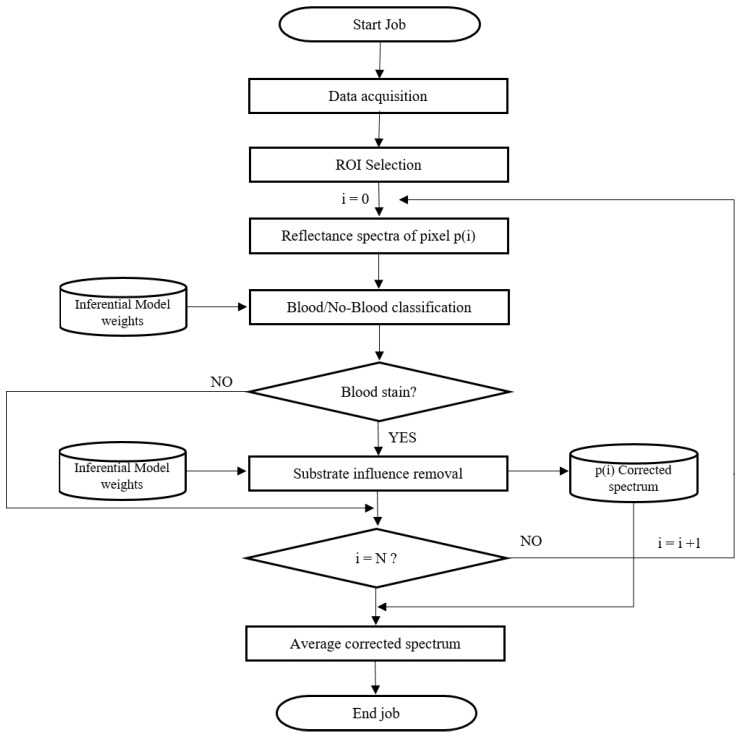
Procedure workflow for the blood stain classification and for the correction of the substrate spectral distortion in the blood stain reflectance spectra.

**Figure 8 sensors-22-07311-f008:**
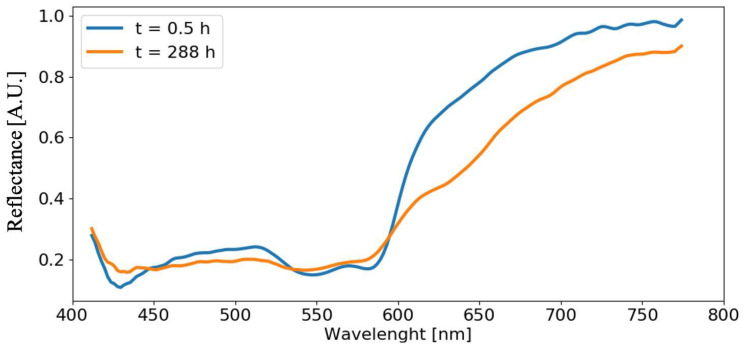
Reflectance spectra comparison between fresh and aged blood on reference substrate $C_{5}$ at time t=0.5 h and t=288 h.

**Figure 9 sensors-22-07311-f009:**
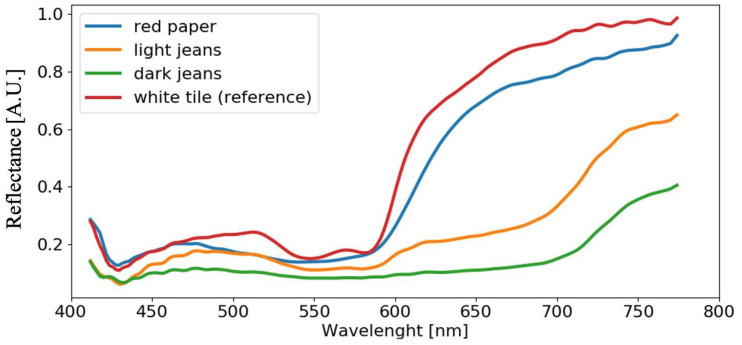
Reflectance spectra comparison on diverse substrates acquired at t=0.5 h.

**Figure 10 sensors-22-07311-f010:**
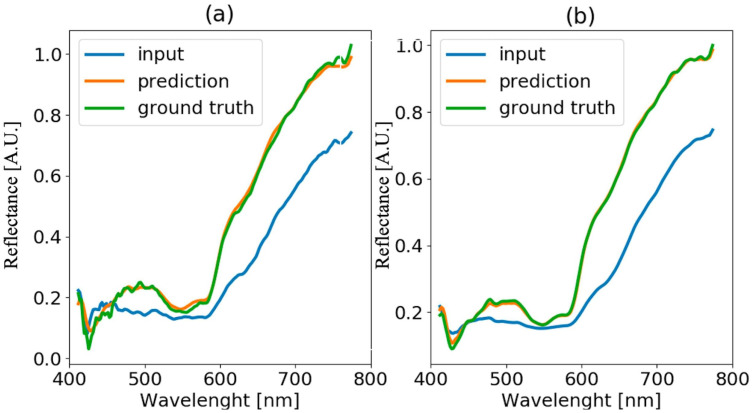
Blood spectra (**a**) and averaged blood spectra (**b**) comparison for the sample on wood substrate and acquired at time t=50 h.

**Figure 11 sensors-22-07311-f011:**
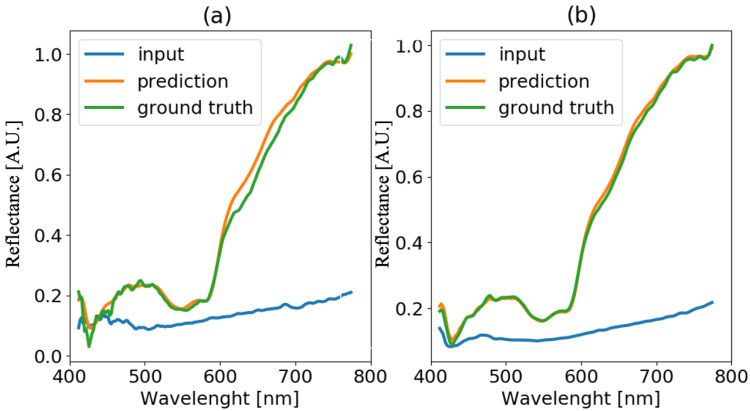
Blood spectra (**a**) and averaged blood spectra (**b**) comparison for the sample on black paper substrate and acquired at time t=50 h.

**Figure 12 sensors-22-07311-f012:**
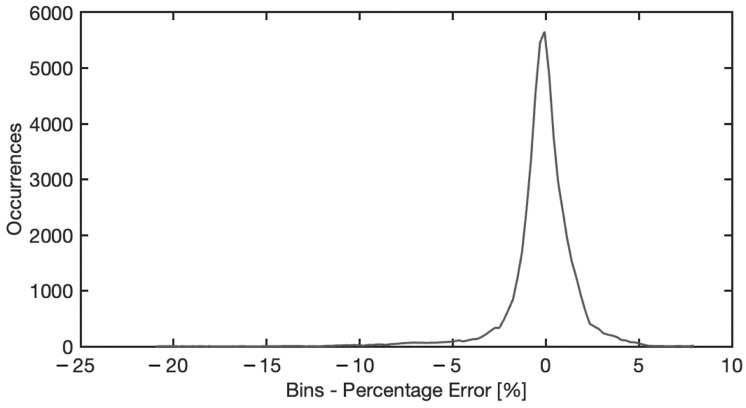
Mean absolute percentage error statistical distribution for the complete set of samples and acquisition times and for the entire spectral range.

**Figure 13 sensors-22-07311-f013:**
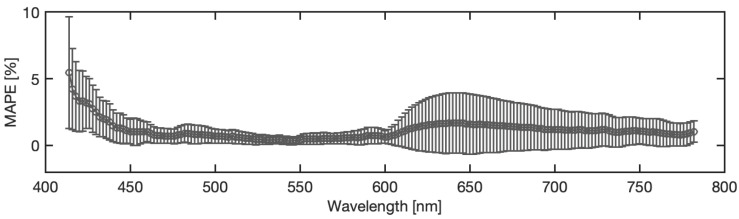
MAPE as a function of wavelength with standard deviation.

**Figure 14 sensors-22-07311-f014:**
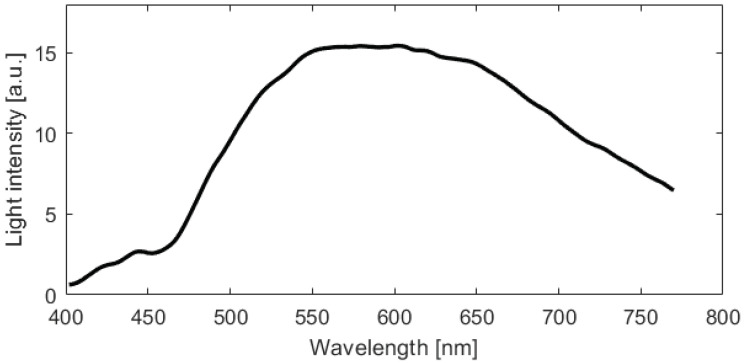
Halogen source intensity spectrum.

**Figure 15 sensors-22-07311-f015:**
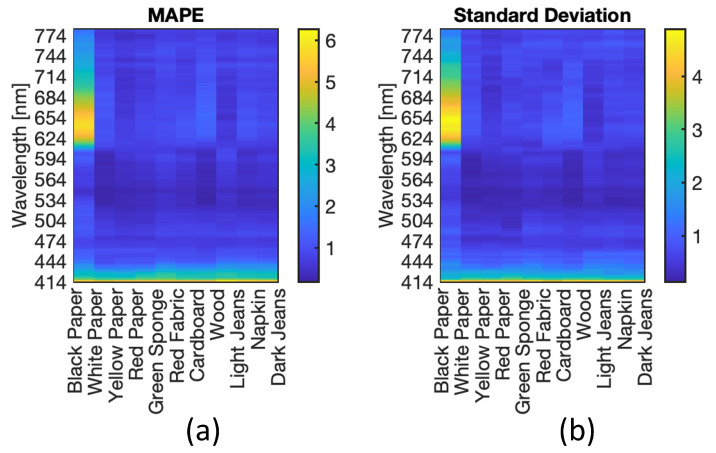
MAPE (**a**) and standard deviation (**b**) as functions of the different substrates and the spectral wavelengths.

**Figure 16 sensors-22-07311-f016:**
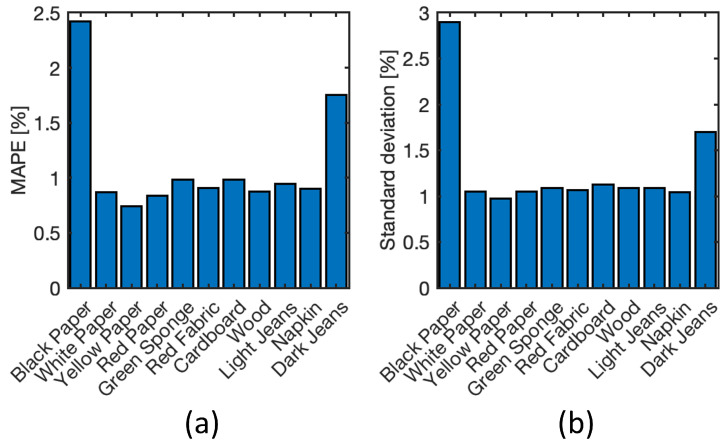
MAPE (**a**) and standard deviation (**b**) of each substrate averaged within the entire spectral range.

**Figure 17 sensors-22-07311-f017:**
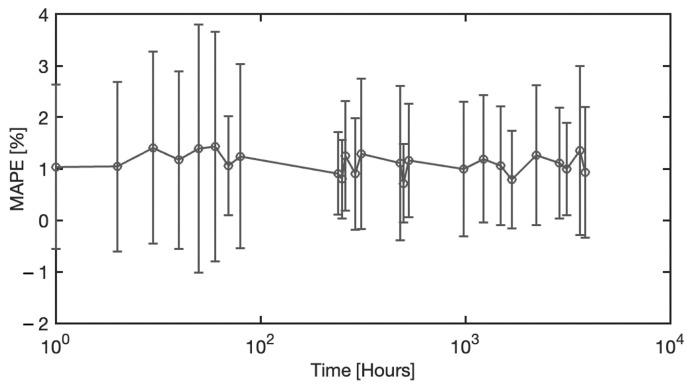
MAPE as a function of the acquisition time, i.e., the age of the blood stain.

**Figure 18 sensors-22-07311-f018:**
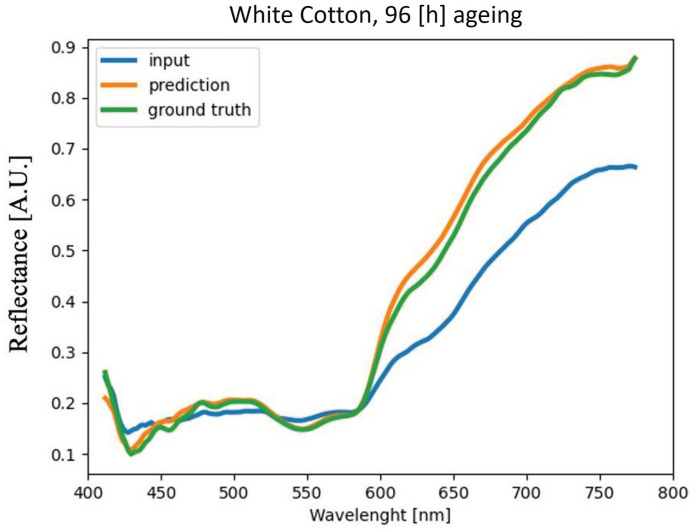
Averaged blood spectra comparison for the sample on white cotton fabric substrate and acquired at time *t* = 96 h.

**Figure 19 sensors-22-07311-f019:**
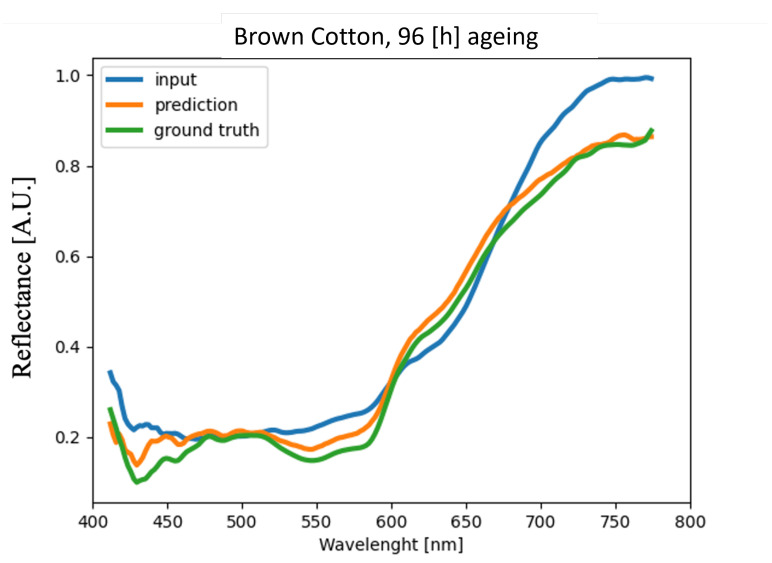
Averaged blood spectra comparison for the sample on brown cotton fabric substrate and acquired at time *t* = 96 h.

**Table 1 sensors-22-07311-t001:** Substrate IDs.

Substrate	ID
Black paper	$C_{1}$
White paper	$C_{2}$
Yellow paper	$C_{3}$
Red paper	$C_{4}$
White ceramic tile	$C_{5}$
Green sponge	$C_{6}$
Red fabric	$C_{7}$
Cardboard	$C_{8}$
Wood	$C_{9}$
Light Jeans	$C_{10}$
Napkin	$C_{11}$
Dark jeans	$C_{12}$

**Table 2 sensors-22-07311-t002:** MAPE in blood stain age determination for blood deposited in two new materials: brown and white cotton fabric.

	MAPE [%]			
	**1 [h]**	**24 [h]**	**47 [h]**	**96 [h]**
White Cotton	2.85%	4.07%	4.35%	4.12%
Brown Cotton	4.88%	3.64%	3.72%	4.91%

## Data Availability

Not applicable.

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
