# Peer review of "Correction of Substrate Spectral Distortion in Hyper-Spectral Imaging by Neural Network for Blood Stain Characterization"

_sensors, 2022, doi:10.3390/s22197311_

Round 1
Reviewer 1 Report
1. Nowadays, deep learning neural networks (DNN) should be able to input images and make a classification in several applications directly. In this paper, the traditional mode of using feature processing and then using ANN as the proposed method should compare its advantages and disadvantages with STOA models.
2. Line 258, [?] cited is missing.
3. Line 272-286, It might be better to make an arrangement, such us items.
4. In Figures 10 and 11, the inputs should not be used to compare with AI’s prediction results. To compare with the results of training, validation, and tests would be sufficient.
5. How many objects are there to be tested? In the case of non-big data, the use of the K-fold method cross-validation will be more able to illustrate the feasibility of the proposed method.
6. The table caption of Table 1 should on the top.
7. It is unclear how many prediction classes are from the output layer. Only two (true or false)? In such cases, DNN is strong, It is recommended to add a multi-class prediction and a comparison with the SOTA model. And also for the regression case study.
8. Eq. (1) should explicitly explain the formula of the objective function.
9. Why is MAPE large only for the black paper?
10. As a result of the conclusion, the model seems overfitting with the limited number of data. There are many ways to prevent this problem via regulation layer, dropout...etc, which is not discussed but might be better to add for prediction enhancement.
Reviewer 2 Report
In this work, the authors consider the problem of correcting the spectral signature of blood stains, which is influenced by the substrate on which it has been deposited.
The work is very interesting, mostly well written (a careful re-reading and minor corrections would be nice), the methodology well described and clear, and the results well demonstrated and convincing.
I would encourage the authors to extend their research, for example, estimating the "age" of the stains, after the correction due to the substrate.
Some minor language issues include
1. The authors use "..an hyperspectral..." while "a hyperspectral" is more appropriate, since the word hyperspectral starts with the sound of a consonant
2. At various locations, the authors use singular form in place of plural, and these minor errors could be corrected. For example "..machine learning approaches has" instead of have
Round 2
Reviewer 1 Report
Thank you for considering my comments.
Minor suggestions for the new lines:
Line 278: "The job..."
Line 278-294: Please make a correction of (1) to (9); it is misleading.
Author Response
We would like to thank the reviewer for the comments and suggestions received. We have made all proposed minor corrections.